# Smartphone-Based Optical Fiber Fluorescence Temperature Sensor

**DOI:** 10.3390/s22249605

**Published:** 2022-12-08

**Authors:** Jianwei Huang, Ting Liu, Yeyu Zhang, Chengsen Zhan, Xiaona Xie, Qing Yu, Dingrong Yi

**Affiliations:** College of Mechanical Engineering and Automation, Huaqiao University, Xiamen 361021, China

**Keywords:** optical fiber, fluorescence, temperature sensor, smartphone

## Abstract

Optical fiber sensors are one preferred solution for temperature sensing, especially for their capability of real-time monitoring and remote detection. However, many of them still suffer from a huge sensing system and complicated signal demodulate process. In order to solve these problems, we propose a smartphone-based optical fiber fluorescence temperature sensor. All the components, including the laser, filter, fiber coupler, batteries, and smartphone, are integrated into a 3D-printed shell, on the side of which there is a fiber flange used for the sensing probe connection. The fluorescence signal of the rhodamine B solution encapsulated in the sensing probe can be captured by the smartphone camera and extracted into the R value and G value by a self-developed smartphone application. The temperature can be quantitatively measured by the calibrated G/R-temperature relation, which can be unified using the same linear relationship in all solid–liquid–gas environments. The performance verifications prove that the sensor can measure temperature in high accuracy, good stability and repeatability, and has a long conservation time for at least 3 months. The proposed sensor not only can measure the temperature for remote and real-time detection needs, but it is also handheld with a small size of 167 mm × 85 mm × 75 mm supporting on-site applications. It is a potential tool in the temperature sensing field.

## 1. Introduction

As one of the most common physical quantities, temperature has a huge impact on the fields of industry, agriculture, biology, medicine and environment; thus, temperature measurement is particularly important. At present, the most common temperature sensors are based on thermoelectric [1,2,3,4] and semiconductor [5] sensors, which have been developed for a long time with mature manufacturing technologies. However, they are susceptible to high voltage and strong electromagnetic interference. In contrast, optical fiber sensors have the advantages of small size, anti-electromagnetic interference, corrosion resistance, and can realize remote detection and real-time monitoring. Therefore, optical fiber temperature sensors have been researched and applied in many fields, which mainly include optical fiber surface plasmon resonance temperature sensors [6,7,8], optical fiber grating temperature sensors [9,10,11], photonic crystal optical fiber temperature sensors [12,13,14], optical fiber Brillouin scattering temperature sensors [15,16,17], and Fabry-Perot optical fiber temperature sensors [18,19,20]. However, some of them suffer from difficulties in the fabrication of sensing probes, such as optical fiber gratings and photonic crystal optical fibers, and others suffer from defects that require special equipment to demodulate signals, such as optical fiber Brillouin scattering temperature sensors and Fabry-Perot optical fiber temperature sensors. Therefore, it is important to develop an optical fiber temperature sensor with a simple sensing probe preparation and simple signal demodulation.

The optical fiber fluorescence temperature sensor based on fluorescence intensity is a technology that combines optical fiber and fluorescence dye to prepare the sensing probe, and uses photodetector to realize fluorescence signal measurement for temperature detection. It is a very simple and effective temperature detection method. Guo et al. [21] prepared the sensing probe by doping NaYF_4_: Yb, Er upconversion nanoparticles (UCNPs) into the polydimethylsiloxane (PDMS) optical fiber, and used the optical fiber spectrometer for signal detection, realizing the human body temperature detection from 25 °C to 70 °C. Yang et al. [22] fabricated the sensing probe by fusing Er^3+^/Yb^3+^ co-doped TeO_2_-ZnO-ZnF_2_-La_2_O_3_ (TZZL) glass with single mode fiber (SMF), and used the optical spectrum analyzer (OSA) to detect the signal, realizing the temperature measurement of the thermal chamber from 30 °C to 287 °C. Ren et al. [23] fabricated the sensing probe by depositing NaBi(WO_4_)_2_: Er^3+^/Yb^3+^ on the end face of the silica fiber, and adopted the optical fiber spectrometer for signal detection, finally realizing the human body temperature measurement of 27–37.6 °C. Tao et al. [24] produced the sensing probe by encapsulating epoxy into the quartz capillary, and used the optical fiber spectrometer for signal detection, and achieved the room temperature measurement from 25 °C to 100 °C. Although the reported optical fiber fluorescence temperature sensors have achieved temperature detection in many different fields, the current fluorescence temperature sensing and measurements are mainly based on the spectrometer or large photodetector, which have disadvantages of large size, high cost and difficulty in on-site detection. Hence, it is of great significance to develop a new optical fiber fluorescence temperature sensor based on a portable and low-cost signal processing device.

Here, we propose a smartphone-based optical fiber fluorescence temperature sensor. The liquid-core sensing probe is prepared by a simple, fast and cost-effective encapsulating method. The sensor uses the smartphone camera for fluorescence signal collection, and a user-friendly smartphone application for image processing and data analysis. All the optical components are integrated into a compact 3D-printed shell with a small size of 167 mm × 85 mm × 75 mm, and the sensing probe can be installed and replaced conveniently by the fiber flange on the shell. The whole system is compact and portable. Moreover, it can realize on-site detection, real-time monitoring and remote measurement. The proposed sensor provides a new solution for the optical fiber temperature detection.

## 2. Sensing Probe Preparation and Sensor Construction

### 2.1. Sensing Probe Preparation

The preparation of the fluorescence sensing probe refers to our previous work [25], as exhibited in Figure 1a. Briefly, a steel capillary with an internal diameter of 0.9 mm, outer diameter of 1.1 mm and length of 80 mm is first immersed into the temperature sensitive solution, which can be filled into the steel capillary through the capillary effect. Then, one end of the capillary is coated with UV curing glue and exposed under a UV lamp for 5 min for sealing. After that, an optical fiber (Scitlion, China) with a core diameter of 105 μm, cladding diameter of 125 μm and protective sleeve diameter of 0.9 mm after end face cleavage is inserted into the other end of the capillary with an insertion ratio of 1/4, and finally sealed also using the UV curing glue after 5 min of UV light exposure. Figure 1b depicts the sensing probe prepared, utilizing the aforementioned procedures. Here, the temperature sensitive solution is Rhodamine B (RhB). Its absorption and emission spectra are shown in Figure 1c, which determines the excitation wavelength of the light source and the passband range of the filter.

### 2.2. Sensor Construction

The schematic of the smartphone-based optical fiber fluorescence temperature sensor used for fluorescence signal excitation and collection is shown in Figure 2a. The reflective-mode system owning a parallel configuration makes it compact and easy for remote and on-site detection. A small 405 nm fiber-coupled semiconductor laser (Haoran Optics, China) with an output power of 3 mW driven by two AA batteries is employed as the excitation light source. A smartphone (Huawei P20 Pro, China) is adopted for fluorescence signal collection and data analysis. A 500 nm long-pass filter (FEL0500, Thorlabs, Newton, NJ, USA) is placed in front of the smartphone camera to reduce the impact of the excitation light. Both the laser and the smartphone camera are connected to the sensing probe via a 2 × 1 fiber coupler with a coupling ratio of 50/50, where the two ports are connected to the laser and smartphone camera, and the one port is connected to the sensing probe. The laser, batteries, filter, fiber coupler, and smartphone are integrated into a handheld holder made of a 3D-printed shell. Additionally, a ferrule connector (FC) fiber flange is present on the shell for simple sensing probe replacement. The sensor is rather compact, only with a size of 167 mm × 85 mm × 75 mm, as shown in Figure 2b,c. It is handheld and powered by inner batteries, indicating that it is a potential temperature measurement tool for on-site application.

### 2.3. Software Design

The wavelength range of the RhB fluorescence emission spectrum is from 500 nm to 600 nm, including the green channel signal (G) and red channel signal (R). A self-referenced signal ratio method of G/R is adopted to measure the temperature, which has a self-calibration ability to reduce the interference caused by environmental factors. The temperature-sensitive fluorescence signal detection and data analysis are implemented on the smartphone. Figure 3a shows the flow chart of the fluorescence signal detection algorithm. First, the image of the fluorescence signal is captured by the smartphone camera, which appears as a spot on a black background. Next, the fluorescence spot is recognized from the captured image based on the threshold segmentation. Then, the fluorescent spot image is divided into an R channel and G channel through RGB color channel separation. Finally, for each channel, the fluorescence intensity is defined as the sum of the fluorescence intensity at each pixel divided by the pixel number within the recognized fluorescence spot.

A user-friendly smartphone application is designed to automatically and quantitatively measure the temperature-sensitive fluorescence signal, as shown in Figure 3b–h. It primarily consists of two functional modules as calibration and detection. It is worth noting that the sensor should be calibrated before detection because the temperature detection is based on the calibration curve of the G/R-temperature relation. Figure 3b depicts the application icon on the smartphone desktop. Figure 3c displays the initial interface. If the calibration has been completed, selecting the “Detection” button opens the detection interface in Figure 3h; if not, clicking the “Calibration” button opens the fluorescence image loading screen in Figure 3d. Pressing the “Load” button opens the camera to take an image or selects one from the album, which then displays on the interface. After entering its corresponding temperature, clicking the “Calculate” button opens the interface in Figure 3e, which shows the fluorescence images of each channel and their calculated intensities, including R, G and G/R. Clicking the “Accept” button records the G/R value and its matching temperature in Figure 3f, and then returns to the fluorescence image loading interface in Figure 3d, while clicking the “Reject” button directly returns to the fluorescence image loading interface in Figure 3d. Clicking the “Calibrate” button in Figure 3e activates the calibration data logging interface in Figure 3f after all the temperatures have been entered and all the G/R values have been calculated. The “Delete” button can be used to remove any data, and the “Back” button can be used to go straight back to the fluorescence image loading interface. By clicking the “Calibration fitting” button in Figure 3f, a quantitative G/R-temperature calibration curve is obtained, as shown in Figure 3g, and the temperature can be measured based on it. The “Back” button in Figure 3g returns back to the calibration data logging interface, and the “Detect” button reaches the detection interface in Figure 3h to quantitatively measure the temperature. Clicking the “Load” button in Figure 3h opens the camera to take an image or load one from the album. Then, clicking the “Detect” button directly displays the corresponding G/R and temperature calculated according to the calibration relationship. The “Back” button takes you to the calibration fitting interface in Figure 3g, and the “Exit” button closes the application. It is worth noting that since the sensor application is developed based on the Android, it can be applied to any Android smartphone. Moreover, since the application contains both calibration and detection, it is convenient to apply it to other smartphones, which only need to be recalibrated and then used for detection.

## 3. Results and Discussion

### 3.1. Sensing Probe Optimization

In order to achieve high-quality temperature sensing, the RhB concentration is optimized. RhB solutions with different concentrations of 0.1 g/L, 0.3 g/L, 0.5 g/L, 0.75 g/L, 1 g/L, 1.5 g/L and 2 g/L are prepared and filled into steel capillaries to fabricate sensing probes. Figure 4a shows the fluorescence images measured at room temperature using the smartphone-based optical fiber temperature sensing system combined with these sensing probes. The R, G, and G/R values corresponding to various RhB concentrations in sensing probes are calculated, as revealed in Figure 4b. It demonstrates that as the RhB concentration increases from 0.1 g/L to 0.3 g/L, both the R and G values increase, and reduce as the RhB concentration increases further. This is mostly because the additional RhB molecules result in an increase in the fluorescence yield at low concentration, which raises fluorescence intensity; meanwhile, more RhB molecules generate aggregation, which decreases the fluorescence intensity [26,27]. Additionally, Figure 4b reveals that the G/R value also increases first and then decreases with the increasing RhB concentration. Since the higher G/R has bigger changes with temperature changes, which makes the sensor have higher sensitivity, the optimal RhB concentration is selected as 0.3 g/L.

### 3.2. Sensor Calibration

In order to verify that the proposed sensor has the ability to quantitatively measure the temperature, the G/R-temperature relations are calibrated in the liquid, solid and gas sensing environments. Liquid environmental calibration is conducted in a water bath, solid environmental calibration is conducted on a constant temperature heating platform (STC803, Germay, China) and gas environmental calibration is conducted in a constant temperature drying oven (DGG-9030B, Senxin, China). The temperature sensing range in all these calibrations is 30 °C to 70 °C with the interval of 4 °C, accounting for the characteristic of RhB. Temperatures are adjusted by different heating devices and measured by extra thermometers. The G/R values under different temperatures are measured by the sensor, and each temperature data is repeated 3 times using different sensing probes. Figure 5a–c plot the G/R-temperature relations for the liquid, solid and gas environments, respectively. As can be observed, all the G/R values are linearly corrected with temperature under various sensing environment conditions, indicating that RhB has the potential to be used for temperature sensing. It is worth noting that all these G/R-temperature relationships are very close, which means that the sensor can be used in all the liquid, solid and gas sample temperature measurements, while only relying on one unified G/R-temperature relation. In order to obtain the unified G/R-temperature relation, it is calibrated according to all the data in Figure 5a–c. As shown in Figure 5d, the sensitivity of sensor is 0.00259/°C. The fitted G/R-temperature relation is used for the quantitative temperature measurement in performance verification and practical applications.

### 3.3. Sensor Performance Verification

The performance of the sensor needs to be verified after completing the calibration, which mainly includes the verification of the accuracy, stability, repeatability and conservation time of the sensor. First, the accuracy of the sensor is tested via measuring temperatures under different sensing environmental conditions. The detected temperatures are within the quantitative temperature sensing range, but different from those used for calibration. Each temperature data is repeated 3 times using three sensing probes. Figure 6a shows that the measured results using the sensor are highly consistent with those obtained by thermometer, which proves that the sensor can measure the temperature in high accuracy. Next, the stability of the sensor is tested by respectively placing three sensing probes in the water bath at 40 °C, 50 °C and 60 °C. The result is recorded every 4 min. As shown in Figure 6b, the measurement result of the sensor can be stable for at least 60 min, which proves that the sensor has a good stability. Furthermore, to test the repeatability and response time of the sensor, the single sensing probe is alternately immersed in two water baths with temperatures of 40 °C and 60 °C. The time interval between two measurements is 1 s. The result in Figure 6c exhibits the good repeatability of the sensor in continuous measurements as well as the fast response time of less than 1 s. Finally, the conservation time of the sensor is tested by immersing the sensing probe conserved at 4 °C for 1, 2 and 3 months into various water baths at 30 °C, 46 °C, 54 °C and 70 °C. The result is shown in Figure 6d. It can be observed that the sensor can still work well in temperature detection after being conserved at 4 °C for 3 months. The above experiments demonstrate that the sensor has good accuracy, stability, repeatability and can be conserved at 4 °C for at least 3 months, which proves that the sensor has an excellent performance in temperature measurement. Furthermore, comparisons of the error bars of the sensor performance verification with the thermometer reveals that the calculation error is within 0.5 °C, indicating that the temperature resolution is 0.5 °C.

### 3.4. Sensor Practical Applications

The good performance of the sensor demonstrates its potential for practical applications. Here, the temperature measurements of the computer surface and hot water-cooling process are used for practical application testing. The temperature of the computer surface can reflect the work state of the computer. For temperature detection, the sensing probe is placed on the bottom of the laptop along with a thermometer. Figure 7a describes the temperature change of the computer surface measured by the sensor with the increase in use time, which well coincides with that measured by the thermometer. For the hot water-cooling process, a sensing probe and a thermometer are directly immersed in the hot water and measure the temperature change process. Figure 7b plots both the sensor and thermometer measured temperatures, and they fit well with each other. The above results prove that the sensor can be well applied to measure temperature in practical applications.

### 3.5. Discussion

As shown in Table 1, compared to existing optical fiber fluorescence temperature sensors, the advantages of the proposed sensor are mainly reflected in two aspects. On the one hand, it is reflected in the preparation method of the sensing probes. The doping method prepares the sensing probes by doping the sensitive materials in the matrix material and combining them with transmission optical fibers. Such sensing probes have stable sensing performance, but their preparation requires expensive instrumentation [28,29,30] or complex chemical processing steps [21]. Physical deposition and chemical modification methods prepare the sensing probes by modifying sensitive materials on the surfaces of optical fibers through electrostatic adsorption or chemical bond coupling. Such sensing probes have a simple and low-cost preparation process, but the sensitive material is easy to fall off, resulting in poor sensing repeatability [31,32,33]. The special optical fiber filling method prepares the sensing probes by filling sensitive materials into the pore structure of special optical fibers. Such sensing probes have high detection sensitivity, but the special optical fiber itself is costly, and the sensitive materials are randomly filled inside the microstructure [34,35,36]. In comparison to these methods, the proposed filling-and-connecting method prepares the sensing probes by filling the capillary tube with sensitive materials and connecting with transmission optical fibers, which is simple, rapid and cost-effective. On the other hand, it is reflected in the signal processing unit. The reported optical fiber fluorescence temperature sensors are usually based on commercial optical fiber spectrometers to achieve sensing, which are large and costly, and are not conducive to field detection. In contrast, the proposed smartphone-based optical fiber fluorescence temperature sensor has the advantages of small size, low cost and being suitable for field detection.

Compared to electrical temperature sensors, the proposed optical fiber fluorescence temperature sensors have the following advantages. First, they can be used for remote detection. Optical fiber has a very low transmission loss, so the sensing probe can flexibly control the length of the transmission optical fiber to achieve remote detection. Meanwhile, electrical temperature sensors generally cannot achieve a very long detection distance. Second, it is free from electromagnetic and voltage interference, so it can be used for detection in harsh environments with strong electromagnetic fields and high voltage. The optical fiber has a core-cladding structure, and the light is transmitted in the fiber core by total reflection, so the external strong electromagnetic fields or high voltage cannot affect the light. Furthermore, the sensing probe can have a very long transmission optical fiber, thus keeping the signal processing unit (the smartphone) away from strong electromagnetic fields or high voltage environments. Meanwhile, electrical temperature sensors are not completely immune to electromagnetic and voltage interference. Third, it can achieve real-time monitoring and in situ analysis. The sensing probe can be placed in the solid–liquid–gas environment to be measured for a long time for testing. Meanwhile, the majority of electrical temperature sensors are designed for a specific application environment.

## 4. Conclusions

In conclusion, a simple, economical, and reliable smartphone-based optical fiber fluorescence temperature sensor is proposed. With a 3D-printed shell, the sensor consisting of a laser, batteries, fiber coupler, filter, and smartphone can be integrated into a handheld POCT device. The liquid-core sensing probe can be prepared by the simple filling-and-connecting method. With a self-developed application, the sensor can perform the temperature calibration and detection functions, and the temperature can be measured by the G/R-temperature relation. It is found that the sensor can use a unified calibration relationship in all solid, liquid and gas environmental conditions. It not only can achieve high-precision temperature measurement, but also has good stability, repeatability and long-term conservation ability. The temperature measurement range of the sensor is 30 to 70 °C with the sensitivity of 0.00259/°C. The response time is within 1 s. The sensing probe has a long conservation time of at least 3 months. The practical applications prove that the sensor has the ability to measure temperature in real life. Considering its low cost, miniaturization, portability and capabilities for real-time monitoring and remote detection, the proposed smartphone-based optical fiber fluorescence temperature sensor has great application potential in temperature sensing fields.

## Figures and Tables

**Figure 1 sensors-22-09605-f001:**
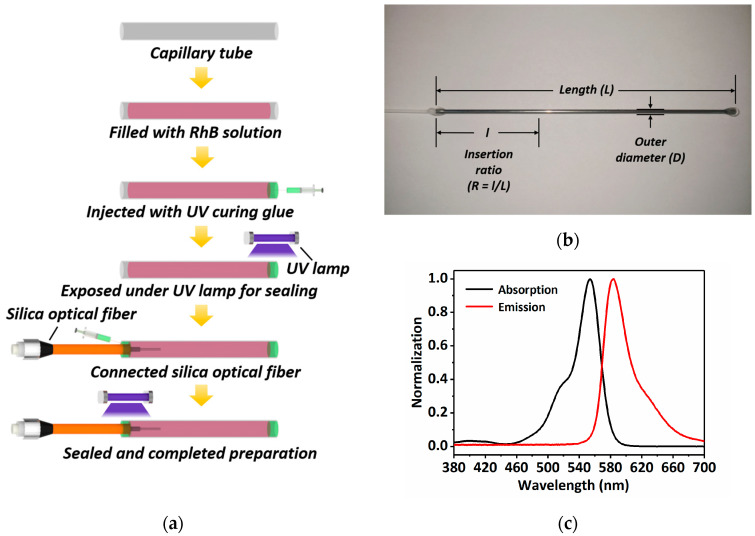
Sensing probe fabrication. (**a**) Sensing probe preparation process; (**b**) prepared sensing probe; (**c**) absorption and emission spectra of RhB.

**Figure 2 sensors-22-09605-f002:**
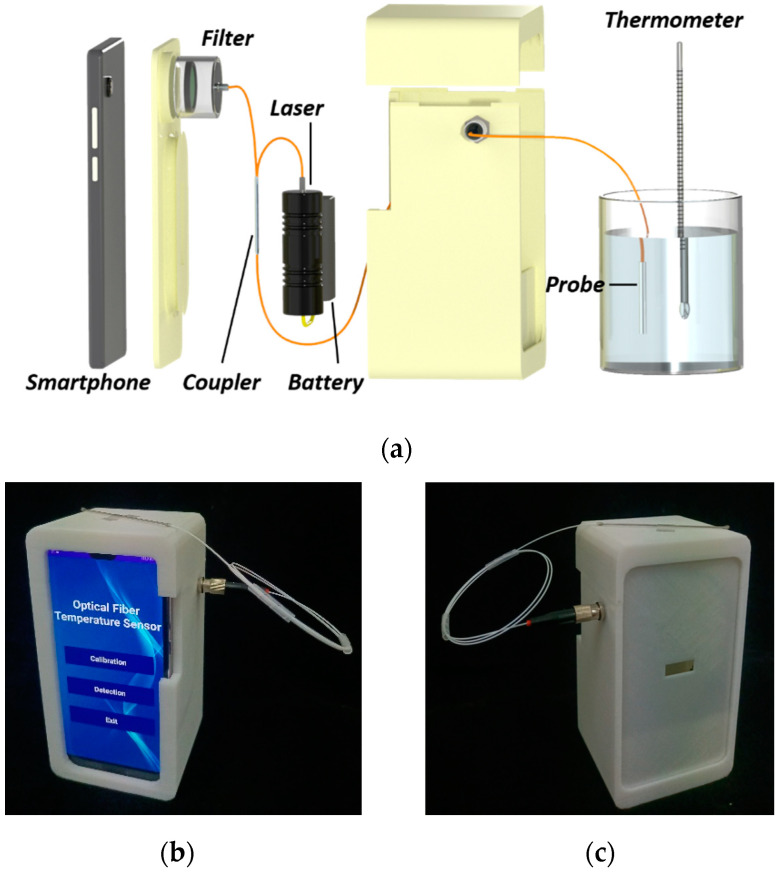
Optical fiber fluorescence temperature sensor. (**a**) Design sketch of the sensor; (**b**) front view photo of sensor; (**c**) back view photo of sensor.

**Figure 3 sensors-22-09605-f003:**
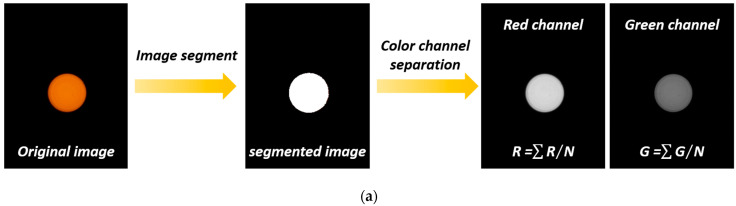
Smartphone application. (**a**) Flow chart of processing of fluorescence image; (**b**) application icon on the smartphone desktop; (**c**) initial interface of application; (**d**) fluorescence image loading interface; (**e**) calculation interface; (**f**) calibration data logging interface; (**g**) calibration result interface; (**h**) detection interface.

**Figure 4 sensors-22-09605-f004:**
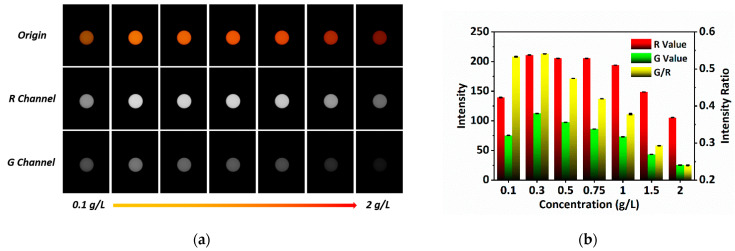
Sensing probe optimization. (**a**) Fluorescence images of sensing probes with different RhB concentrations; (**b**) R value, G value and G/R of sensing probes with different RhB concentrations.

**Figure 5 sensors-22-09605-f005:**
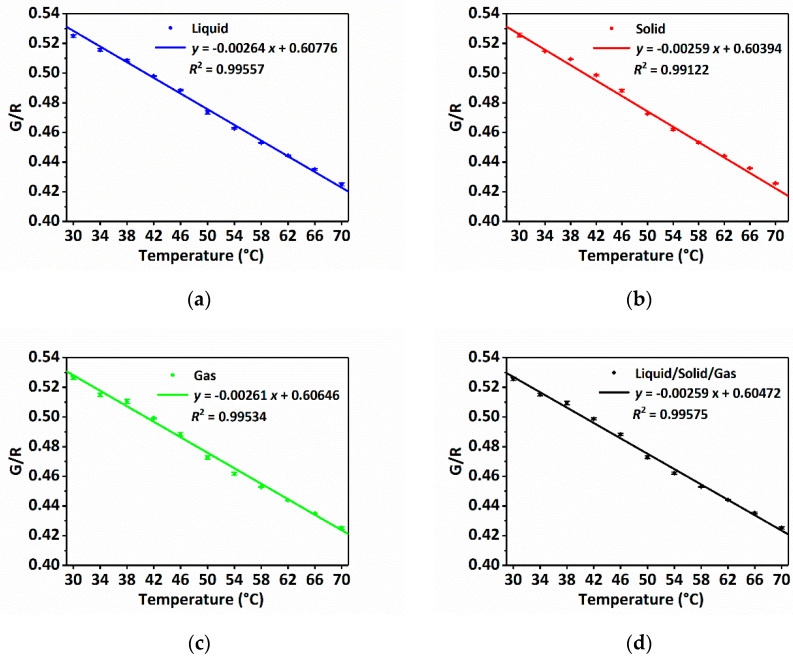
Sensor calibration. (**a**) Measured and fitted G/R-temperature relation in the liquid sensing condition; (**b**) measured and fitted G/R-temperature relation in the solid sensing condition; (**c**) measured and fitted G/R-temperature relation in the gas sensing condition; (**d**) the unified G/R-temperature relation for all environmental conditions.

**Figure 6 sensors-22-09605-f006:**
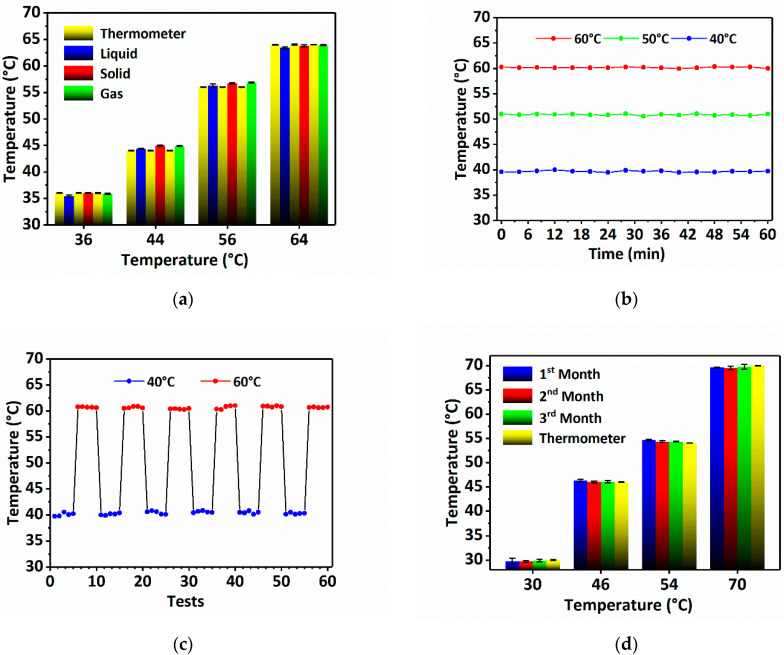
Sensor performance verification. (**a**) Accuracy verification; (**b**) stability verification; (**c**) repeatability and time response verification; (**d**) conservation time verification.

**Figure 7 sensors-22-09605-f007:**
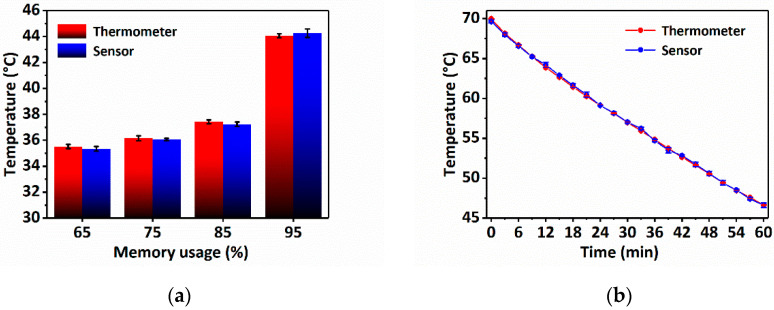
Sensor practical applications. (**a**) Laptop temperature detection; (**b**) hot water-cooling temperature detection.

**Table 1 sensors-22-09605-t001:** Comparison of different optical fiber fluorescence temperature sensors.

Sensing Probe Preparation Method	Sensing Probe Preparation Complexity	Signal Processing Unit	Sensor Cost	Refs
Doping method	Normal	Spectrometer	High	[21,28,29,30]
Physical deposition and chemical modification method	Normal	Spectrometer	High	[31,32,33]
Special optical fiber filling method	Complex	Spectrometer /optical spectrum analyzer	High	[34,35,36]
Filling-and-connecting method	Easy	Smartphone	Low	This work

## Data Availability

Not applicable.

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
