# Peer review of "Smartphone-Based Optical Fiber Fluorescence Temperature Sensor"

_sensors, 2022, doi:10.3390/s22249605_

Round 1

Reviewer 1 Report

This paper demonstrated a smartphone based optical fiber fluorescence temperature sensor, which shows advantages of low-cost, real-time detection and smart. I would recommand its publication provided that the following comments are taken into account: 

1. When compared with thermometers, the error should be calculated quantitatively.

2.  A comparsion with other temperature sensors is suggested to add as a table.

3. During the performance verifications and practical application, the quantitative estimations, such as the measuring errors should be added.

Author Response

Reviewer 1:

Comments:

This paper demonstrated a smartphone based optical fiber fluorescence temperature sensor, which shows advantages of low-cost, real-time detection and smart. I would recommand its publication provided that the following comments are taken into account:

Thanks for your suggestions. We have revised the manuscript according to your comments and answered the questions below. We hope these modifications can satisfy your requirements.

  1. When compared with thermometers, the error should be calculated quantitatively.

Compared with the thermometer, the calculation error is within 0.5 °C. We have added the calculation error in the revised manuscript.

  1. A comparison with other temperature sensors is suggested to add as a table.

According to your suggestion, we have added a table about comparisons with other temperature sensors.

Temperature sensor type

Preparation

Cost

Signal process

Optical fiber fluorescence sensor

Simple

Low

Easy

Optical fiber surface plasmon resonance sensor

Complex

Normal

Normal

Optical fiber grating sensor

Complex

High

Normal

Photonic crystal optical fiber temperature sensors

Complex

High

Normal

Optical fiber Brillouin scattering temperature sensors

Normal

Low

Complex

  1. During the performance verifications and practical application, the quantitative estimations, such as the measuring errors should be added.

According to your suggestion, we have added the measuring errors in the performance verifications and practical applications.

We hope these modifications can satisfy your requirements. Thanks again for your advices and suggestions on our research!

Reviewer 2 Report

The manuscript proposes a temperature sensing system that utilizes the fluorescence spectra of Rhodamine molecules for monitoring. Unlike traditional fiber-optic methods, the activity of chemical molecules decays over time, and while the manuscript claims to achieve long conservation time for at least 3 months, it is not competitive for fiber-based temperature sensors that can be stored almost permanently. In terms of sensing performance indicators, themanuscript does not give specific parameters such as the temperature resolution and response speed, which will affect the sensor performance in practical applications. For example, some occasions require a high refresh rate for temperature measurements, but can the fluorescence spectra of chemical molecules in the manuscript have a high response speed? Moreover, data processing using smartphones is not a novel method for sensing demodulation. For example, in the work of [1], a smartphone was used to analyze and demodulate the signal, and to measure the concentration of ethylene gas. Therefore, for the structure of the fluorescence spectrum of chemical molecules as transducers, the authors should focus on the sensing fields such as biochemical molecule detection, but not competitive for physical quantities such as temperature and strain.

Reference:

[1] Yang, Xin, et al. "Quantitative Determination of Ethylene Using a Smartphone-Based Optical Fiber Sensor (SOFS) Coupled with Pyrene-Tagged Grubbs Catalyst." Biosensors 12.5 (2022): 316.

Author Response

Reviewer 2:

Comments:

The manuscript proposes a temperature sensing system that utilizes the fluorescence spectra of Rhodamine molecules for monitoring. Unlike traditional fiber-optic methods, the activity of chemical molecules decays over time, and while the manuscript claims to achieve long conservation time for at least 3 months, it is not competitive for fiber-based temperature sensors that can be stored almost permanently.

The fluorescence-based optical fiber temperature sensors do suffer from the problem that the activity of chemical molecules decays over time, which is an inevitable problem for all fluorescence-based optical fiber temperature sensors. Although the traditional optical fiber temperature sensors (such as fiber grating sensors, fiber Brillouin scattering temperature sensors) can be stored almost permanently, they still have some shortcomings, such as requiring complex signal demodulation equipment, being difficult to miniaturize, and cannot be used for on-site detection. On the contrary, the fluorescence-based optical fiber temperature sensor has the advantages of simple sensing probe preparation, convenient signal demodulation, and easy miniaturization. From these perspectives, the fluorescence-based optical fiber temperature sensor is competitive.

In terms of sensing performance indicators, the manuscript does not give specific parameters such as the temperature resolution and response speed, which will affect the sensor performance in practical applications.

Through the error bars, the temperature resolution is 0.5 °C; and through Figure 6c, the response time is within 1 s. We have added these in the revised manuscript.

Moreover, data processing using smartphones is not a novel method for sensing demodulation. For example, in the work of [1], a smartphone was used to analyze and demodulate the signal, and to measure the concentration of ethylene gas.

Reference: [1] Yang, Xin, et al. "Quantitative Determination of Ethylene Using a Smartphone-Based Optical Fiber Sensor (SOFS) Coupled with Pyrene-Tagged Grubbs Catalyst." Biosensors 12.5 (2022): 316.

Although data processing using smartphones for sensing demodulation has been reported, the self-referenced signal ratio method of G/R based on the same spectrum used for temperature measurement proposed in this manuscript can effectively reduce the effect of environmental interference on the signal. This is innovative.

Therefore, for the structure of the fluorescence spectrum of chemical molecules as transducers, the authors should focus on the sensing fields such as biochemical molecule detection, but not competitive for physical quantities such as temperature and strain.

The fluorescence spectrum of Rhodamine B can change with temperature, so it has the potential to be used for temperature sensing. Furthermore, using it for temperature sensing has many advantages, such as the fact that the sensing probe preparation is simple and low cost, the signal demodulation is simple, the system is easy to miniaturize for on-site detection, and so on. Our subsequent research may focus on biochemical molecule detection.

Thanks for your suggestions. We have revised the manuscript according to your comments and answered the questions below. We hope these modifications can satisfy your requirements.

Reviewer 3 Report

The authors propose a smartphone based optical fiber fluorescence temperature sensor. They claim that all the components including the laser, filter, fiber coupler, batteries, and smartphone are integrated into a 3D printed shell, on the side of which there is a fiber flange used for sensing probe connection. The fluorescence signal of rhodamine B solution encapsulated in the sensing probe can be captured by the smartphone camera and extracted into R value and G value by a self-developed smartphone application. I liked the manuscript and how the title sounds but I would mention some revisions that need to be corrected before the publication.

1. Fig.2a does not clearly describe the setup. Looks like the coupler's common port passes through the laser. Please check this.

2. You have used Huawei P20 Pro to detect the signal. Does your system design allow to use other smartphones, having other cameras, OS, and, sure, different dimetions and cameras position? If not, the advantage of the smartphone use turns to the disadvantage, I think...

3. For now there are a lot of smartphones having integrated temperature-vision feature, without using the probes and other installations. The advantages of your sustem in a comparison with this approach need to be highlighted in the paper.

4. I feel lack of technical parameters in the conclusion section. If you present the sensor, its general characteristics should be given. The most important of them is accuracy.

5. Some of your sections end with the drawings, please place them within text, after the first mentioning.

Author Response

Reviewer 3:

Comments:

The authors propose a smartphone based optical fiber fluorescence temperature sensor. They claim that all the components including the laser, filter, fiber coupler, batteries, and smartphone are integrated into a 3D printed shell, on the side of which there is a fiber flange used for sensing probe connection. The fluorescence signal of rhodamine B solution encapsulated in the sensing probe can be captured by the smartphone camera and extracted into R value and G value by a self-developed smartphone application. I liked the manuscript and how the title sounds but I would mention some revisions that need to be corrected before the publication.

Thanks for your suggestions. We have revised the manuscript according to your comments and answered the questions below. We hope these modifications can satisfy your requirements.

  1. Fig.2a does not clearly describe the setup. Looks like the coupler's common port passes through the laser. Please check this.

According to your suggestion, we have updated Fig. 2a. In addition, we have added the explanation in the revised manuscript that the two ports of the 2×1 fiber coupler are connected to the laser and smartphone camera, and the one port is connected to the sensing probe.

  1. You have used Huawei P20 Pro to detect the signal. Does your system design allow to use other smartphones, having other cameras, OS, and, sure, different dimetions and cameras position? If not, the advantage of the smartphone use turns to the disadvantage, I think...

Since the sensor application is developed based on Android, it can be applied to any Android smartphone. And since the application contains both calibration and detection, it is convenient to apply it to other smartphones, which only need to be recalibrated and then used for detection.

If the smartphone has different dimensions and camera position, the shell of the 3D structure needs to be re-printed, which is the characteristic of all smartphone-based sensing systems.

We have added these in the revised manuscript.

  1. For now there are a lot of smartphones having integrated temperature-vision feature, without using the probes and other installations. The advantages of your system in a comparison with this approach need to be highlighted in the paper.

The smartphones integrated with the temperature-vision feature can only be used to detect the temperature rise inside the smartphone, not the external temperature.

Our proposed sensor can not only realize remote detection, real-time monitoring, and on-site measurement, but also measure temperatures of solid, liquid and gas samples using a unified calibration relationship with high accuracy and good repeatability. We have mentioned these in the conclusion section.

  1. I feel lack of technical parameters in the conclusion section. If you present the sensor, its general characteristics should be given. The most important of them is accuracy.

According to your suggestion, we have added the technical parameters in the conclusion section. Such as the temperature measurement range of the sensor is 30 to 70 °C with the sensitivity of 0.00259 /°C. The response time is within 1 s. The sensing probe has a long conservation time of at least 3 months.

  1. Some of your sections end with the drawings, please place them within text, after the first mentioning.

According to your suggestion, we have adjusted the figures to the appropriate positions in the revised manuscript.

We hope these modifications can satisfy your requirements. Thanks again for your advices and suggestions on our research!
